# Stochastic Volatility Models with Skewness Selection

**DOI:** 10.3390/e26020142

**Published:** 2024-02-06

**Authors:** Igor Martins, Hedibert Freitas Lopes

**Affiliations:** Insper Institute of Education and Research, Rua Quatá 300, São Paulo 04546-042, Brazil; hedibertfl@insper.edu.br

**Keywords:** stochastic volatility, sparsity, skewness

## Abstract

This paper expands traditional stochastic volatility models by allowing for time-varying skewness without imposing it. While dynamic asymmetry may capture the likely direction of future asset returns, it comes at the risk of leading to overparameterization. Our proposed approach mitigates this concern by leveraging sparsity-inducing priors to automatically select the skewness parameter as dynamic, static or zero in a data-driven framework. We consider two empirical applications. First, in a bond yield application, dynamic skewness captures interest rate cycles of monetary easing and tightening and is partially explained by central banks’ mandates. In a currency modeling framework, our model indicates no skewness in the carry factor after accounting for stochastic volatility. This supports the idea of carry crashes resulting from volatility surges instead of dynamic skewness.

## 1. Introduction

Accurate representation of asset returns is one of the key topics in finance. Based on the theoretical results from [1,2], standard approaches to asset pricing have largely focused on the first and second moments. Stochastic volatility (SV) models, discussed, for example, by [3], are among the cornerstone models in modern financial econometrics. In their simplest form, SV models represent asset returns via normal distribution with persistent volatility and a mean that is either constant or a linear function of explanatory variables. Such models capture the first two moments of asset returns in a simple and elegant manner, being supported empirically and theoretically, as discussed in [4].

While we acknowledge the importance of the first two moments in asset pricing, we also recognize the potential benefits of including skewness when modeling returns. Due to its ability to capture the likely direction of returns, models with time-varying skewness may be more suitable for forecasting periods with a higher concentration of same sign returns, leading to better detection of both overperformance and underperformance periods. Ref. [5] is one key example of the empirical benefits of adding such feature for modeling cross-sectional stock momentum. While the momentum factor is known for delivering good mean–variance compensation, it is also subject to a long period of negative performance. By capturing such prolonged periods of likely negative returns via dynamic skewness, ref. [5] improves the performance of the stock momentum factor compared to traditional approaches that neglect skewness.

However, including dynamic skewness in traditional financial econometric models can be costly. While allowing for asymmetry may lead to a better representation of some financial time series, it may not be a vital feature, and its inclusion risks overparameterization. Therefore, we wish to include dynamic skewness only when required by the data and remove such a feature if it is not necessary.

This paper expands stochastic volatility models by allowing dynamic skewness without having to impose it. We replace the traditional hypothesis of Gaussian errors with a skew-normal distribution. Such a change preserves the usual features for the first two moments of SV models but allows for dynamic skewness. Since the inclusion of time-varying asymmetry may not always be necessary, we consider a sparsity-inducing scheme for its parameters. In particular, we consider a random-walk evolution for the asymmetry. When the standard deviation of the dynamic process is shrunk to zero, our model results in an SV model with constant skewness. Additionally, if the level of skewness is shrunk to zero, we recover a traditional SV model. By combining prior information with the likelihood of the model, our proposed approach automatically chooses between dynamic, static, or no skewness.

We consider two empirical applications. In our first application, we model Brazil and US bonds, obtaining three main results. First, our proposed model indicates that bond yield changes for both countries, better represented by including time-varying skewness, resulting in out-of-sample improvements for forecasting the direction of future yields when compared to generalized autoregressive score (GAS) models. Second, the recovered skewness is associated with interest rate cycles of monetary easing and tightening. Third, inflation and unemployment partially explain the recovered skewness, linking it to central banks’ mandates. In a second application, we model the carry factor for currency returns. Our model indicates not only the lack of dynamics, but also no skewness at all for it after accounting for volatility. Therefore, similar to the crash mitigation via volatility scaling proposed in [6], the carry factor also experiences reduced crashes once dynamic volatility is considered, without requiring the inclusion of skewness.

This paper intersects and contributes to multiple areas. First, it contributes to the stochastic volatility literature by expanding the static skewness model of [7] to the dynamic case while also extending the sparsity-inducing scheme of [8] by allowing for dynamic skewness to shrink towards the static case. Second, it contributes to the toolbox of methods for recovering dynamic skewness. Ref. [9] relies on option data, while [10] uses rolling windows. Both approaches have limitations. While theoretically sound, option-based approaches require tradeable options with a large collection of strikes, high liquidity, and continuous expiration dates. Such requirements are hard to meet in many applications, especially for emerging markets such as in our Brazilian bond applications. Rolling window-based approaches artificially introduce dynamics into skewness by changing the sample period continually. Such changes come at the cost of outliers playing a large role in estimation, in addition to a trade-off based on window size, which affects the precision of the estimate and the speed at which the dynamics change. Third, it contributes to the interest rate literature by showing that inflation and unemployment partially explain skewness, linking it to central banks’ mandates, and expanding the traditional mean and variance analysis of papers such as [11,12]. Fourth, it contributes to the debate of whether the carry factor presents dynamic skewness after accounting for heteroskedasticity, as discussed in [13,14], by claiming that skewness is unlikely to be dynamic and, in fact, is more likely to be zero after accounting for SV effects.

The paper is organized as follows. It starts by describing traditional SV models and moves on to our proposal with time-varying skewness in Section 2. Section 3 discusses the sparsity-inducing framework. Section 4 shows our Hmailtonian Monte Carlo (HMC) approach to simulate from the joint posterior. Section 5 presents the bond yield forecasting application while Section 6 shows the carry factor application. Section 7 concludes.

## 2. SV Model with Time-Varying Skewness

A traditional stochastic volatility model is defined in Equations (Equation 1)–(Equation 5). Equation (Equation 1) represents asset returns with a mean of zero and dynamic volatility of $exp(h_{t}/2)$. Equation (Equation 2) indicates persistent log-volatility with level μ and persistence ϕ, having initial values given by the stationary distribution shown in Equation (Equation 3). In its simplest form, both measure and state equations have Gaussian errors as represented in Equations (Equation 4) and (Equation 5).
(1)$$y_{t}=exp\left(\frac{h_{t}}{2}\right)\varepsilon_{t},$$
(2)$$h_{t}=\mu_{h}+\phi_{h}\left(h_{t-1}-\mu_{h}\right)+\sigma_{h}\eta_{t},$$
(3)$$h_{0}∼N\left(\mu_{h},\frac{\sigma_{h}^{2}}{1-\phi_{h}^{2}}\right),$$
(4)$$\varepsilon_{t}∼N\left(0,1\right),$$
(5)$$\eta_{t}∼N\left(0,1\right).$$

In order to introduce asymmetry into the model, we replace the normal variable, $\varepsilon_{t}$ in Equation (Equation 4) with a skew-normal random variable, $z_{t}$. We utilize the skew-normal density representation described in [15,16] as shown in Equation (Equation 6), where ϕ(·) and Φ(·) represent standard normal density and distribution function, respectively. λ controls the degree of asymmetry in the distribution, as depicted in Figure 1. Specifically, for λ=0, the skew-normal reduces to standard normal distribution. Additionally, we can introduce location (ξ) and scale (ω) parameters to the skew-normal, denoting X=ξ+ωz as X∼SN(ξ,ω,λ), with its density presented in Equation (Equation 7). Appendix A describes the relationships between ξ, λ, and ω in terms of mean, variance, and skewness.
(6)IfZ∼SN(λ),thenp(z;λ)=2ϕ(z)Φ(λz);
(7)$$\mathrm{If}\;X∼SN\left(\xi ,\omega ,\lambda \right),\;\mathrm{then}\;p\left(x;\xi ,\omega ,\lambda \right)=2\frac{1}{\omega }\phi \left(\frac{x-\xi }{\omega }\right)\Phi \left(\lambda \frac{x-\xi }{\omega }\right).$$

We allow for the possibility of dynamic skewness by replacing static λ with $\lambda_{t}$, which evolves according to a random walk starting at $\lambda_{0}$ as represented by Equations (Equation 8) and (Equation 9). Therefore, we modify our observation equation by enabling both volatility and skewness to vary with time, changing the observation equation to Equation (Equation 11). While Equation (Equation 1) implies that $y_{t}$ follows a symmetric distribution, Equation (Equation 11) indicates that $y_{t}$ comes from a potentially skewed distribution. Therefore, our model can be viewed as an extension of the stochastic volatility models with static skewness proposed in [7,8,17].
(8)$$z_{t}∼SN\left(\lambda_{t}\right),$$
(9)$$\lambda_{t}=\lambda_{t-1}+\sigma_{\lambda }\eta_{\lambda ,t}\;\mathrm{with}\;\eta_{\lambda ,t}∼N\left(0,1\right),$$
(10)$$\lambda_{0}=\alpha_{0},$$
(11)$$y_{t}=exp\left(\frac{h_{t}}{2}\right)z_{t}.$$

Since $\varepsilon_{t}∼N\left(0,1\right)$, Equation (Equation 1) implies that $y_{t}$ follows a normal distribution with a location of zero and a scale of $exp(h_{t}/2)$. By replacing $\varepsilon_{t}$ with $z_{t}∼SN\left(\lambda_{t}\right)$, Equation (Equation 11) implies that $y_{t}$ follows a skew-normal distribution with a location of zero, a scale of $exp(h_{t}/2)$, and skewness parameter $\lambda_{t}$.

We are not the first to consider models with dynamic skewness. For instance, refs. [18,19] introduced GAS models which induce time-varying skewness through the score of conditional density. More recently, refs. [5,20] used GAS models to model Bitcoin returns and manage momentum-based portfolios. However, unlike our proposal, GAS models are suitable for including time-varying higher moments within a GARCH-like framework, while our proposal is designed for SV models.

## 3. Sparsity-Inducing Approach

The inclusion of asymmetry may improve the representation of time series by capturing periods with a higher concentration of returns exhibiting the same sign, as demonstrated, for instance, in [5,16,21]. However, it may not always be necessary as it involves estimating additional parameters. To mitigate the risk of overparametrization, our aim is to incorporate time-varying skewness only when required by the data. One possible approach involves estimating multiple models with different skewness specifications and then selecting a model using the Bayes factor. Ref. [22] provides a comprehensive review of using Bayes factors for model selection. However, recovering Bayes factors can be a challenging problem, as highlighted in [23]. We propose a sparsity-inducing method that conveniently performs model selection without the need to estimate multiple models and entirely avoids the use of Bayes factors.

In the non-Bayesian literature, shrinkage is based on maximizing the likelihood of a model subject to a penalty function with the LASSO of [24] being the most commonly employed approach. From a Bayesian perspective, shrinkage problems can be framed as a penalization of the log-likelihood through a log-prior. In fact, the posterior mode of a linear model with Double Exponential prior having a location of zero and a scale of 2/ψ equals the point estimate of the LASSO with penalty ψ, as shown in [25]. Therefore, to shrink both $\sigma_{\lambda }$ and $\lambda_{0}$ towards zero, we utilize Double Exponential priors for both parameters, as illustrated in Equations (Equation 12) and (Equation 13).

Our proposal is consistent with the concept of sparsity in dynamic models, a principle also utilized in [26,27,28,29]. Our approach is considered sparsity-inducing in the sense that both the random walk variance and initial state for the skewness parameter are shrunk towards zero. Equation (Equation 9) governs the dynamics of the asymmetry parameter. When $\sigma_{\lambda }$ approaches zero, $\lambda_{t}$ becomes static and assumes the value of its initial point, $\lambda_{0}$. Furthermore, if the initial point is also zero, the model reduces to the vanilla SV model. Hence, by inducing sparsity for both $\sigma_{\lambda }$ and $\lambda_{0}$, we can encompass all three cases of interest.
(12)$$\sigma_{\lambda }∼\mathrm{DoubleExponential}\left(0,1/\kappa_{\sigma_{\lambda }}\right)\;\mathrm{with}\;\kappa_{\sigma_{\lambda }}∼Gamma\left(a,b\right),$$
(13)$$\alpha_{\lambda }∼\mathrm{DoubleExponential}\left(0,1/\kappa_{\alpha }\right)\;\mathrm{with}\;\kappa_{\alpha }∼Gamma\left(c,d\right).$$

To the best of our knowledge, we are the first to introduce sparsity in the dynamic skewness framework. The closest paper to ours is [8], which uses the spike and slab prior of [30] to estimate the posterior probability of inclusion of static skewness.

## 4. Remaining Priors and Posterior Inference

Our modeling approach involves the following unknown quantities: $\Theta =\mu_{h},\phi_{h},\sigma_{h},\alpha_{0},\sigma_{\lambda },h_{t},\lambda_{t}$. Our goal is to estimate the joint posterior distribution p(Θ|y), which, according to Bayes’ rule, can be calculated as
$$p\left(\Theta |y\right)=\frac{p(y|\Theta )p(\Theta )}{\int p(y|\Theta )p(\Theta )d\Theta }.$$
p(y|Θ) is the likelihood component being characterized by our proposed sampling model. We consider independent components for each member of p(Θ). $p(\mu_{h})$ and $p(\phi_{h})$ follow Gaussian distributions, $p(\sigma_{h}^{2})$ has inverse gamma distribution while $p(\alpha_{0})$ and $p(\sigma_{\lambda })$ have double exponential distribution, as discussed previously. The exact values for the prior parameters are described in Appendix B and Appendix C.

After defining p(y|Θ) and p(Θ), we must recover p(Θ|y). However, ∫p(y|Θ)p(Θ)dΘ does not have an analytical solution. Moreover, due to the high dimensionality of Θ, grid-based integration methods are computationally infeasible. Thus, we employ to Markov Chain Monte Carlo (MCMC) methods to sample from p(Θ|y). In this paper, we use a particular MCMC method, known as HMC, instead of the more traditional Random–Walk Metropolis Hastings (RWMH).

Both HMC and RWMH are MCMC methods. Thus, both methods generate a proposed value for unknown quantities $\Theta^{\left(i\right)}$ on each Markov chain iteration i in order to approximate p(Θ|y). We denote the sequence of $\Theta^{\left(i\right)}$ by ${\left\{\Theta^{\left(i\right)}\right\}}_{i=1}^{I}$. In our applications, we run our HMC scheme for 30,000 iterations with the first 15,000 being used as burn-in draws. HMC and RWMH differ in their approach to generating $\Theta^{\left(i\right)}$ from $\Theta^{\left(i-1\right)}$. RWMH generates $\Theta^{\left(i\right)}$ as a random walk from the previously sampled $\Theta^{i-1}$. In contrast, HMC enhances the process by using guided proposals based on the gradient of the log posterior. This guidance helps direct the Markov chain toward regions of higher posterior density while also sampling tail areas properly, as discussed in [31,32]. Specifically, HMC generates proposals by solving the following Hamiltonian equations:$$\frac{dq}{dt}=-\frac{\partial H(\Theta ,q)}{\partial \Theta }=\nabla_{\Theta }logp\left(\Theta |y\right),$$
$$\frac{d\Theta }{dt}=\frac{\partial H(\Theta ,q)}{\partial q}=\frac{\partial K(q)}{\partial q}=M^{-1}q,$$
where H(Θ,q) represents the Hamilton function. In the context of this paper, $H\left(\Theta ,q\right)=-logp\left(\Theta |y\right)+\frac{1}{2}q^{T}M^{-1}p$. Additionally, $\nabla_{\Theta }logp\left(\Theta |y\right)$ is the gradient of the log posterior density. Due to $\nabla_{\Theta }logp\left(\Theta |y\right)$, ref. [31] claims that HMC generates guided proposals. We refer readers to [31,32] for a thorough review of HMC methods.

We choose HMC over RWMH for two main reasons. First, due to its more refined method to generate $\Theta^{\left(i\right)}$, HMC requires several iterations less than RWMH. This difference becomes even bigger if Θ is high-dimensional which is the case in this paper. Second, the software Stan verison 1.2, introduced in [33], offers convenient implementation of HMC. In our study, we utilize Stan in conjunction with R through the rStan package introduced in [34] (The codes and data for this project are available on the first author’s GitHub page, github.com/igorfbmartins, shortly after publication).

## 5. Empirical Application: Bond Yields

The bond market is one of the largest in the world being key for investors and policymakers. Most papers focus on the first two moments, e.g., refs. [11,12,35,36]. Our paper focuses on the much less explored third moment. Skewness captures the likely direction of returns allowing for an interest rate investor to improve their forecast about the sign of future yields. Such feature is explored in the literature in at least three ways. First, ref. [5] highlights the role of skewness in forecasting crashes in momentum portfolios. Second, works [37,38] highlight that even monetary ’surprises,’ such as the differences between actual policy rates announced by the Federal Open Market Committee (FOMC) and expectations from professional forecasters, can be partially predicted by the option-implied skewness of the 10-year US bond. Third, ref. [17] demonstrates that allowing for skewness improves value-at-risk evaluation for multiple assets.

In our first application, we model the monthly yield changes in fixed 1-year maturity for both American and Brazilian bonds. We sample from the joint posterior via the HMC scheme presented in Section 3 by combining the likelihood implied by our proposed model in Section 2 and the priors described in Appendix B. In both cases, skewness is likely to be time-varying. It is associated with cycles of monetary easing and tightening. It is partially explained by the central bank’s mandate and provides valuable information regarding the future direction of yield changes.

The sample of US bonds comes from the updated dataset of [39] made available by the Federal Reserve Board starting on July 1981 and lasting until August 2023. The sample of Brazilian bonds is based on the DI interest rate contracts available on the Brazilian stock and future exchange B3. The DI contracts are interpolated to form 1-year fixed maturity bonds using the same Nelson–Siegel procedure described in [39] resulting in a sample starting in February 2004 and ending in August 2023. Figure 2 plots both time series of monthly yield changes. Notably, both series fluctuate around zero with volatility clusters. Both series hint at the possibility of skewness. For example, the US series presents a persistent and negative yield change period in early 1990 and early 2000. In the Brazilian series, the negative and persistent yield changes are even clearer, with the period from August 2016 to March 2018 being one example.

Some might express concern that the guidance provided by central banks regarding future rates entirely dictates yield changes. However, such concerns are unlikely to be substantial for several reasons. First, if central bank communication were the sole determinant of yields for 1-year fixed maturity bonds, yields would instantly adjust to incorporate the information from newly announced paths and persist at that level until subsequent announcements, in line with the efficient market hypothesis. Second, current yields not only reflect the anticipated average future short rate over a bond’s lifespan, but also encompass a term premium component. As demonstrated, for instance, in [40,41], this term premium component is likely to be time-varying. Consequently, yields are expected to fluctuate over time even after accounting for the expected future short rate. Third, while central bank communication indeed aids in forecasting future yields, it is by no means the sole source of information, as exemplified in [42].

We apply our proposed model to both US and Brazilian bonds with the priors presented in Appendix B. To account for the Brazilian time series being shorter than the American, the estimation sample comes closer to the present than the American. Precisely, we consider an estimation period which lasts up to December 2018 for the US and up to December 2019 to the Brazilian case. Table 1 presents the posterior summaries for the parameters of both time series. The values of $\phi_{h}$ indicate high persistence in the log-scale which is reflected in Figure 3 that plots posterior summaries of $\left\{h_{t}\right\}$. In both cases, our approach captures the surges in volatility indicated in Figure 2. For instance, in the US series, our model captures spikes in volatility during the early 2000s and the financial crisis of 2008. Similarly, the Brazilian time series shows a volatility spike in 2008, which our model also captures. Additionally, both series are likely to have a dynamic skewness as shown by the posterior summaries of $\sigma_{\lambda }$ with the Brazilian bonds having a bigger range of variation for $\lambda_{t}$ than bonds from the US. This evidence is supported by the posterior summaries of $\left\{\lambda_{t}\right\}$ in Figure 4 as well.

Figure 5 plots the posterior mean of $\left\{\lambda_{t}\right\}$ alongside monetary easing–tightening cycles. The green shaded areas represent periods of monetary easing characterized by interest rate cuts implemented by the countries’ central banks. Conversely, the red shaded areas represent periods of monetary tightening marked by interest rate hikes. Our approach identifies negative skewness in yield changes during easing periods and large positive skewness when a central bank is hiking interest rates. The monetary policy cycles for the US are based on the FED effective fund rate, while for the Brazilian case, they are derived from the target rate policy decisions of each meeting (Brazilian target policy rate is available at https://www.bcb.gov.br/controleinflacao/historicotaxasjuros (accessed on 28 November 2023)).

If the asymmetry of yield changes is connected to interest rate cycles, then drivers of the monetary policy should help explain skewness. We test this hypothesis by regressing the posterior mean of $\left\{\lambda_{t}\right\}$, denoted $\hat{\lambda }$, into inflation and unemployment as represented by Equation (Equation 14). Both variables are reasonable ex-ante since central banks have mandates of price stability and full employment.
(14)$${\hat{\lambda }}_{t}∼N\left(X\beta ,\sigma^{2}\right)\;\mathrm{with}\;X=\left[1,Inflation_{t},Unemployment_{t}\right].$$

Table 2 indicates that unemployment levels partially explain the asymmetry in yield changes for both countries. The negative sign of $\beta_{unemployment}$ is reasonable since central banks typically address high unemployment levels by implementing interest rate cuts to encourage consumption, and, as shown in Figure 5, easing cycles are associated with negative skewness on yield changes. Moreover, in the case of Brazil, inflation is likely to also influence this asymmetry. The positive sign of $\beta_{inflation}$ aligns with central banks’ tendency to counter inflation surges by raising interest rates. As illustrated in Figure 5, this is associated with higher values of skewness. Regarding the US, although inflation posed challenges in the late 1970s and early 1980s, for most of our estimation sample, the American economy did not face significant inflationary pressures. Consequently, yields did not react substantially to inflation. Therefore, the nearly zero effect of $\beta_{\lambda }$ is justifiable.

We also evaluate the out-of-sample performance of our proposed model. If skewness is informative about the likely direction of changes in bond yields, then $sign(E_{t}{\hat{\lambda }}_{t+1})$ should agree with $sign(y_{t+1})$. We verify this claim within an increasing window framework, where the initial window corresponds to the estimation sample. For each window, we execute our HMC procedure and obtain $sign(E_{t}{\hat{\lambda }}_{t+1})$. Table 3 presents our findings. For US bonds, we observe that $sign(y_{t+1})$ matches $sign(E_{t}\lambda_{t+1})$ in 66.1% of cases, with an average change of 20.6% when correctly forecasted and only 8.7% when incorrect. Similarly, in the case of Brazilian bonds, $sign(y_{t+1})$ aligns with $sign(E_{t}\left[\lambda_{t+1}\right])$ in 72.7% of instances. Moreover, the average magnitude of correctly predicted yield changes is 8.2%, while it is only 2.4% when the prediction is incorrect. Therefore, out-of-sample analysis supports our assertion that dynamic skewness in bond yield changes serves as a predictive variable for future yield changes. Furthermore, we compare our results with skewness-based forecasts using GAS models via the implementation of [43] (check https://cran.r-project.org/web/packages/GAS/ (accessed on 28 November 2023)). While our proposal demonstrates similar performance to those of GAS models in the US case, it outperforms GAS in the Brazilian case. Thus, for our dataset, our proposed model not only predicts the future direction of yield changes, but also shows improvements over GAS models.

## 6. Empirical Application: Carry Factor

In addition to the bond yield application detailed in Section 5, we explore application in the currency market. According to [44], two factors—dollar factor and carry—account for the cross-section of currency returns. Our focus is on the latter factor, which captures interest rate differentials between countries. It involves taking long positions on countries with high interest rate differentials relative to the US and short positions on countries with the smallest differentials. While our emphasis is on the FX markets, ref. [45] argues that carry-based factors can effectively explain the cross-section of various asset classes, including commodities and equities. For this analysis, we consider and update a sample of the carry factor described in [44] (readers can check out Lustig’s carry factor on gsb-faculty.stanford.edu/hanno-lustig/files/2022/05/CurrencyPortfolios.xls), presented in Figure 6, which starts on November 1983 and lasts up to May 2021.

We are not the first to study the skewness of carry returns. For example, refs. [10,13] identify a time-varying crash risk on the carry factor. This risk materializes in some occasions leading to large negative returns to the carry factor and skewing its distribution to the left. Conversely, ref. [14] uses out-of-the-money currency options hedging against large crashes and shows that carry remains profitable, indicating a small role for tail risk on currency returns. Additionally, ref. [14] presents evidence in favor of time varying volatility for carry returns and that the largest negative return of its sample, October 2008, occurs in a period of high volatility. Our model is well suited to evaluate such claims. Ref. [14] is not an isolated case. Ref. [6] provides another example of tail risk mitigation in tradeable factors after accounting for volatility without skewness playing a major role.

We consider the prior specification shown in Appendix C and sample from the posterior using the HMC scheme described in Section 4. Figure 7 plots the posterior mean (black), interquartile range (green) and q05-q95 interval (red) for both $\left\{exp\left(\frac{h_{t}}{2}\right)\right\}$ and $\left\{\lambda_{t}\right\}$ which are shown at the top and bottom panel, respectively. The top panel corroborates the evidence of time-varying volatility with a surge in volatility around October 2008 similarly to the description in [14]. The bottom panel indicates that it is likely that there is no skewness at all after accounting for stochastic volatility.

Additionally, we assess the performance of the carry factor during periods of high volatility compared to low volatility. We say that volatility is high if it is above the average volatility recovered for the full sample and we say it is low otherwise. Table 4 reports the results of our analysis. The average return for the carry factor is the same in both environments. However, crash indicators such as the return on the fifth percentile and minimum return exhibit significant improvements. Therefore, in combination with the results shown in Figure 7, our results indicate that after accounting for volatility, it is unlikely that skewness plays a large role in affecting the returns of the carry factor.

## 7. Conclusions

This paper expands stochastic volatility models by allowing for time-varying skewness without having to impose it. By considering a LASSO-type regularization for both the standard deviation and the starting level of the skewness dynamics, our model selects among dynamic, static, and no skewness in a data-driven approach. In our bond yield application, we highlight the benefits of dynamic skewness by demonstrating its connection to monetary easing/tightening cycles and central banks’ mandates. Additionally, we show that asymmetry provides information about the likely direction of future bond yield changes. In the second application, we shed light on the debate of carry average returns reflecting time-varying skewness versus no skewness but time-varying volatility. Our model indicates no skewness after accounting for stochastic volatility.

## Figures and Tables

**Figure 1 entropy-26-00142-f001:**
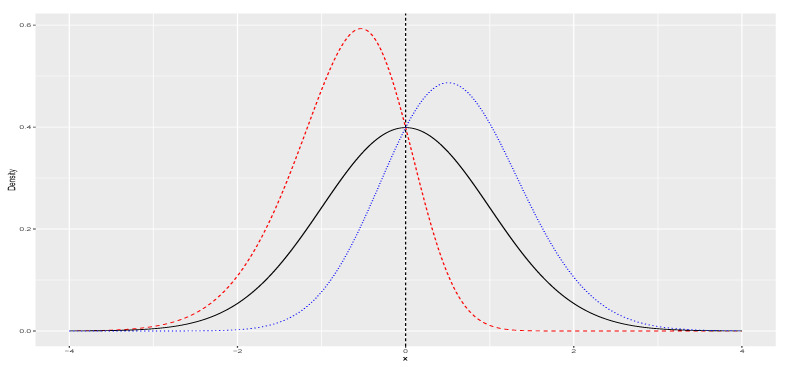
Skew-normal densities with ξ=0 and ω=1 and λ equal to −2, 0, and 1 for the dashed red, solid black and dotted blue lines, respectively. If λ<0, the distribution is skewed to the left. If λ=0, SN(0) reduces to the standard Gaussian distribution. Finally, if λ>0, the distribution is skewed to the right.

**Figure 2 entropy-26-00142-f002:**
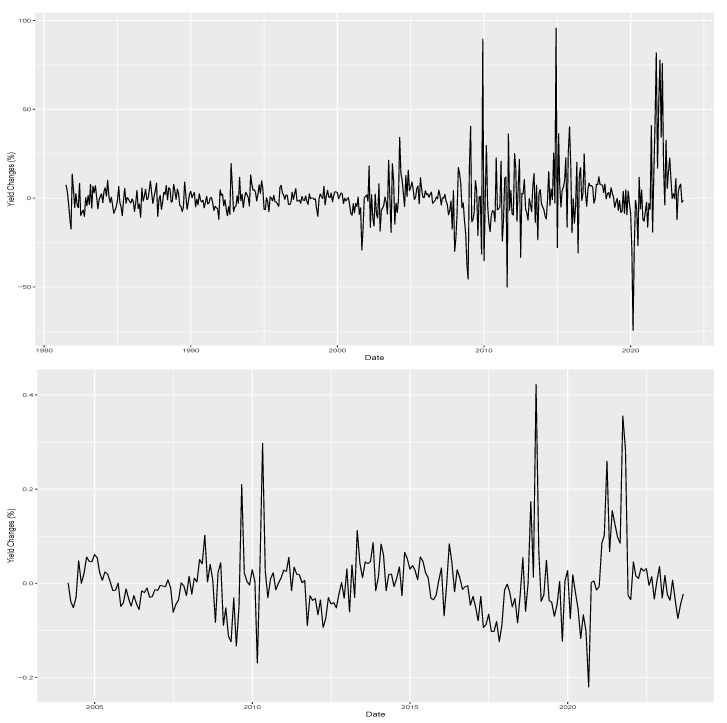
Change in yields from fixed 1-year maturity bonds from the US and Brazil. The top panel presents monthly changes in yields from 1-year bonds from the US government starting in July 1981 and lasting up to August 2023. Similarly, the bottom panel shows monthly changes in 1-year Brazilian bonds from February 2004 up to August 2023.

**Figure 3 entropy-26-00142-f003:**
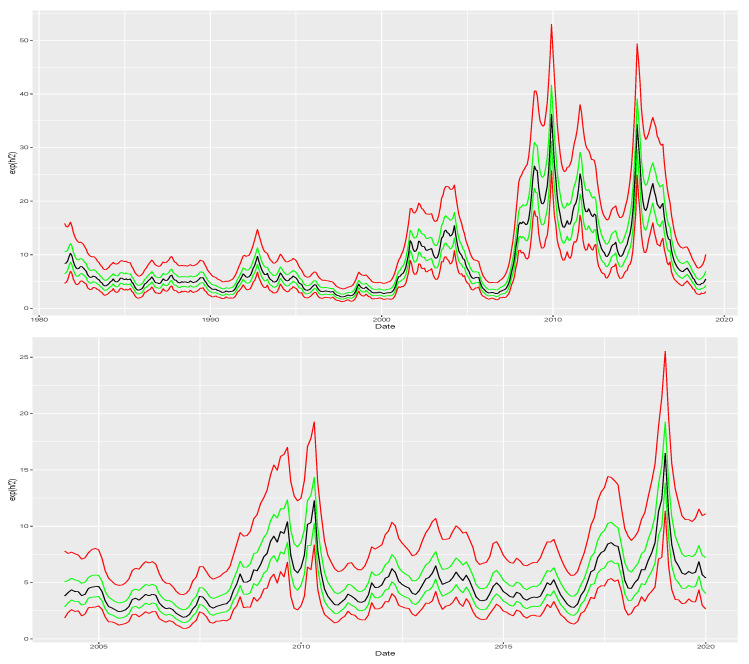
Scale recovered for the US and Brazilian bonds during the estimation period which lasts up to December 2018 for the US and up to December 2019 for the Brazilian case. Green lines represent quantiles 0.25 and 0.75 while solid red lines represent quantiles 0.05 and 0.95. For both series, time-varying volatility is plausible. Top panel indicates rises in yield change volatility during the early 2000s and around the global financial crisis for the US, while the bottom panel indicate peaks of volatility for Brazilian yield changes between 2009 to 2010 with a second spike in late 2018 and early 2019.

**Figure 4 entropy-26-00142-f004:**
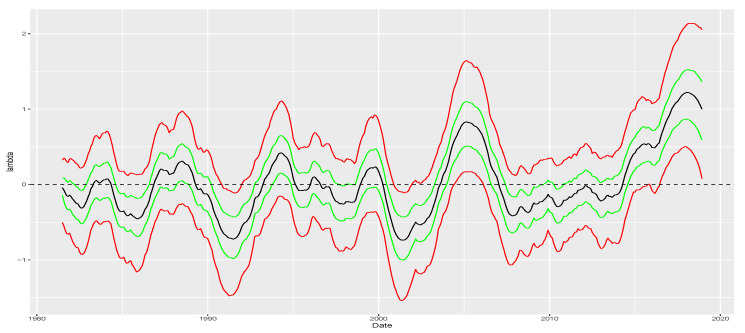
$\left\{\lambda_{t}\right\}$ recovered for the US and Brazilian bonds during the estimation period which lasts up to December 2018 for the US and up to December 2019 for the Brazilian case. While λ is not the skewness itself, it can be transformed by using the formula in Appendix A. Green lines represent quantiles 0.25 and 0.75 while solid red lines represent quantiles 0.05 and 0.95. Time-varying skewness is likely in both cases. We obtain stronger evidence of dynamic skewness for Brazil in the bottom panel than for the US in the top one.

**Figure 5 entropy-26-00142-f005:**
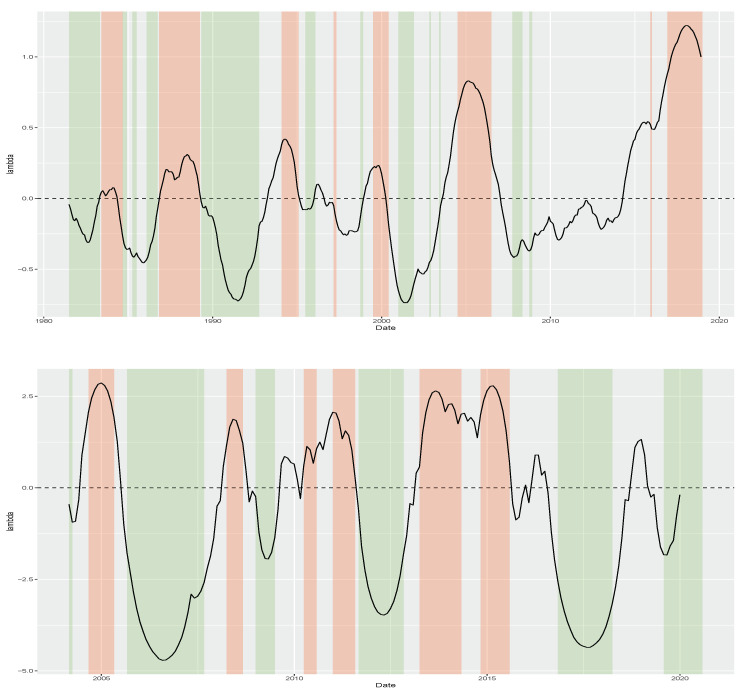
$\left\{\lambda_{t}\right\}$ recovered for the US, top panel, and Brazilian bonds, bottom panel, during the estimation period which lasts up to December 2018 for the US and up to December 2019 for the Brazilian case. Green shaded area indicate monetary easing periods while red ones indicate monetary tightening. In both cases, $\lambda_{t}$ seems able to capture the monetary cycles and reflect future direction of yield change. US monetary police cycles are based on the FED effective fund rate, while for the Brazilian case they are recovered from the target rate policy rate decision of each meeting.

**Figure 6 entropy-26-00142-f006:**
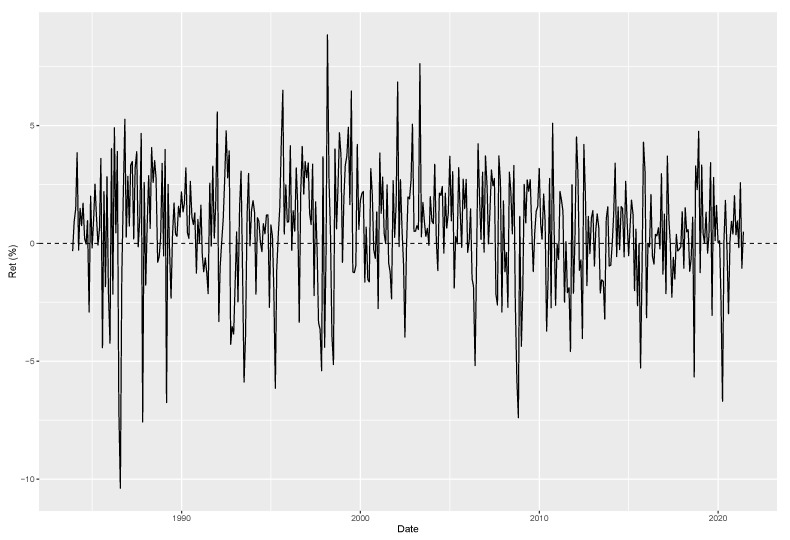
Updated version of the time series for the Carry factor introduced in [44]. Our sample lasts from November 1983 to May 2021.

**Figure 7 entropy-26-00142-f007:**
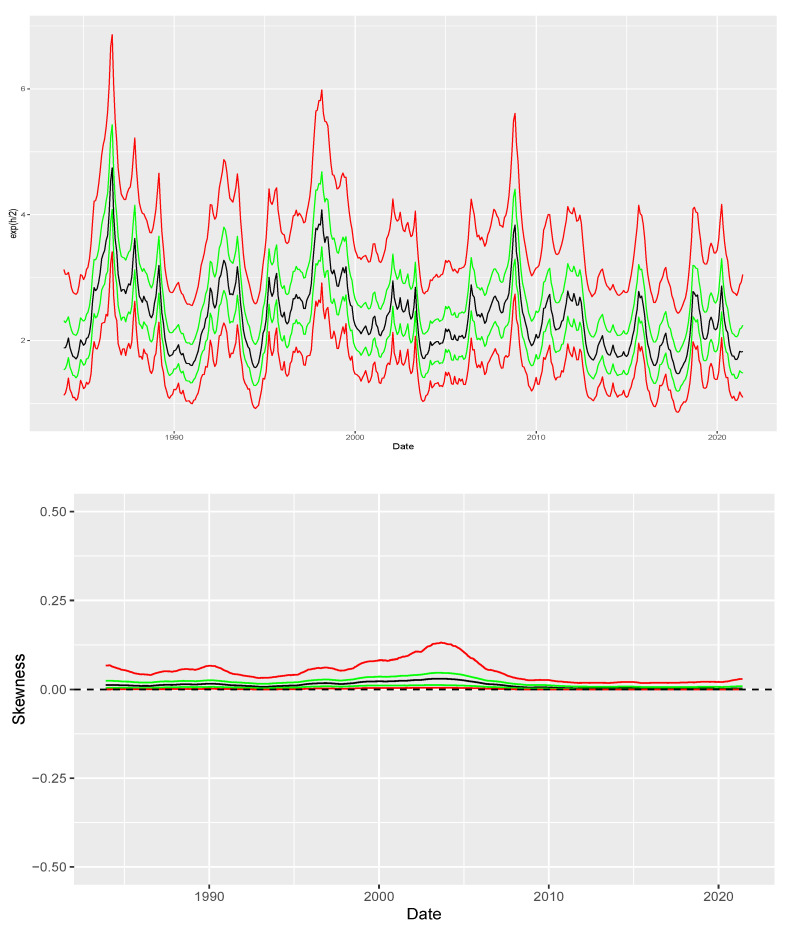
Skewness recovered from the carry factor. Green lines represent quantiles 0.25 and 0.75 while solid red lines represent quantiles 0.05 and 0.95. After controlling for volatility, we find no evidence of skewness in the carry factor.

**Table 1 entropy-26-00142-t001:** Posterior summaries of the parameters from our proposed model. $\mu_{h}$, $\phi_{h}$ and $\sigma_{h}$ represent the log-scale level, persistence and standard deviation, respectively. $\sigma_{\lambda }$ captures the dynamic of the skewness component and $\alpha_{0}$ is the initial level of the skewness. Both series are likely to have a dynamic skewness as shown by the posterior summaries of $\sigma_{\lambda }$ with the Brazilian bonds having a bigger range of skewness variation than bonds from the US.

	$\mu_{h}$	$\phi_{h}$	$\sigma_{h}$	$\sigma_{\lambda }$	$\alpha_{0}$
US q05	2.94	0.92	0.32	0.09	−0.62
US Mean	3.88	0.96	0.42	0.17	−0.06
US q95	4.84	0.99	0.55	0.27	0.42
BR q05	2.59	0.59	0.38	0.63	−2.67
BR Mean	3.08	0.79	0.63	1.09	−0.46
BR q95	3.59	0.94	0.91	1.75	0.79

**Table 2 entropy-26-00142-t002:** Linear regression of the posterior mean of $\left\{\lambda_{t}\right\}$ into inflation and unemployment during the estimation sample. For both countries, unemployment partially explains the skewness in yield changes, while the asymmetry in the Brazilian case is also partially explained by inflation.

	Intercept	$\beta_{\mathrm{Inflation}}$	$\beta_{\mathrm{Unemployment}}$
US q05	0.39	−0.01	−0.11
US mean	0.58	0.00	−0.09
US q95	0.73	0.01	−0.06
BR q05	−0.72	1.16	−0.35
BR mean	0.39	2.47	−0.23
BR q95	1.57	3.52	−0.09

**Table 3 entropy-26-00142-t003:** If bond yield skewness captures the likely direction of yield changes, then we should expected yield increases when $E_{t}[\lambda_{t+1]}>0$ and, otherwise, decreases. Our model provides evidence supporting such claims. In both cases, our model correctly predicts the correct yield change direction in at least 66.1% of times.

	Proposal US (%)	GAS US (%)	Proposal Brazil (%)	GAS Brazil (%)
Hit Ratio	66.1	64.3	72.7	54.5
Avg when right	20.6	20.7	8.2	7.8
Avg when wrong	8.7	8.8	2.4	5.2

**Table 4 entropy-26-00142-t004:** Summary statistics of the carry factor in high- and low-volatility environments for the sample starting in November 1983 and lasting up to December 2021. While the average return for the carry factor in both environments is the same, crash indicators such as the return on the 5th percentile and minimum return are severely improved.

	High Vol	Low Vol
Mean	0.58	0.57
Sd	3.34	1.41
Q05	−5.38	−2.07
Min	−10.38	−3.21

## Data Availability

Data used in this project is available at the first author’s GitHub page: github.com/igorfbmartins.

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
