# Peer review of "Stochastic Volatility Models with Skewness Selection"

_entropy, 2024, doi:10.3390/e26020142_

Round 1

Reviewer 1 Report

Comments and Suggestions for Authors

The paper is co-authored by one of the top experts in Bayesian methods with one of his PhD students. The paper is great in its present form, and I am happy to recommend its publication.

Author Response

Dear, reviewer. 

Thank you for your comments and for recommending our paper to be published.

Reviewer 2 Report

Comments and Suggestions for Authors

This interesting paper expands traditional stochastic volatility models by allowing for time-varying skewness without imposing it. While dynamic asymmetry may capture the likely direction of future asset returns, it comes at the risk of leading to overparameterization. This proposed approach mitigates this concern by cleverly leveraging sparsity-inducing priors to automatically selects the skewness parameter as being dynamic, static or zero in a data-driven framework. Two empirical  and interesting applications are considered. 

The paper is interesting and I recommend this paper for the publication

Author Response

(The authors gave the same response as above.)

Reviewer 3 Report

Comments and Suggestions for Authors

Thank you for the opportunity to review this paper, I found it very ineresting even though it is slightly out of my main area of research. The paper is generally well written, and it represents a nice contribution to literature on SV models. 

I have no major issues or concerns, but the language and style are not great, so I'm afraid some things might have gotten "lost in translation". It took me two read-through to figure out the main ideas of the paper, and that is due to the language barrier. I believe the authors should consult an editor or native speaker to express their ideas more clearly, especially in 1) explaining what sparsity means in this context and 2) in Section 4. First, sparsity has a meaning in the context of image and signal processing, and I can't see how that would apply here. Also, I am not greatly familiar with RWMH and HMC, Sec. 4 was barely legible to me and I couldn't understand what the procedure was, lines 157-170 in particular (what is the "acceptance ratio"?).

Other minor comments:

- Fig. 1: lines could perhaps have different types (dashed etc.) so that they can be distinguished even in black/white print. Also a legend on the figure would help.

- What is the difference  between Eq. 1 and 11? h_t is modeled differently, but the equation is the same, so what is changed?

- Bayes factors are used to compare hypotheses, not models. Perhaps the authors meant "Bayesian information criterion" (BIC)?

Comments on the Quality of English Language

Due to time constraints, below are only corrections up to (not including ) Sec. 5. It is also possible that I missed something, so I would highly recommend detailed proof-reading.

11: selects -> select

14: an -> a

45: gaussian -> Gaussian

64: have -> has

72: which -> with(?)

73: several -> many/most(?)

74 (and elsewhere): specially -> especially

74: when is extra

86: Section3 missing space

87: acronym HMC should be explained when first introduced

102: insert "density" before representation

112: to -> and

126: multiples -> multiple

173: main... what? responsible

177: comes -> come

183: introduce -> introduced

Author Response

Dear, reviewer.

Thank you for your comments. We addressed all your points and made several changes to the original text in order to improve the paper.

Main comments:

1) What sparsity means in this context:
Our paper uses the concept of sparsity in dynamic models which is also used in Frühwirth-Schnatter and Wagner [2010], Belmonte et al. [2014], Bitto and Frühwirth-Schnatter [2019] and Lopes et al. [2022]. In DLMs, we must recover the path of all coefficients instead of a single point. Usually, such path is assumed to follow a random-walk (RW). In the DLM with RW dynamic, we have two quantities of interest: the variance of the random walk and the initial state.
If the variance of the random walk is zero, then the dynamic model reduces to a static model. In particular, Frühwirth-Schnatter and Wagner (2010) show that the variance selection problem for DLMs is equivalent to a variable selection problem. Therefore, if the variance each component of a vector of time-varying coefficients is equal to zero, we have sparsity on the variance component of the DLM. Similarly, if we shrunk the initial states to zero, we have sparsity of the initial conditions.
In our paper, sparsity has a similar meaning to the DLM case. The skewness parameter λt follows a random walk. We allow for both the variance of λt (represented by σλ ) and the initial state (represented by λ0 ) to be shrunk
towards zero. Therefore, we refer to it as a sparsity inducing approach. Note that if σλ = 0, our model reduces to a SV with static skewness. Additionally, if λ0 is also 0, then our model reduces to a traditional SV without skewness.
We changed Section 3 to better explain the meaning of sparsity in our paper by including the following explanation:
” Our proposal is consistent with the concept of sparsity in dynamic models, a principle also utilized in Frühwirth-Schnatter and Wagner [2010], Belmonte et al. [2014], Bitto and Frühwirth-Schnatter [2019], and Lopes et al. [2022].
Our approach is considered sparsity-inducing in the sense that both the random walk variance and initial state for the skewness parameter are shrunk towards zero. Equation (9) governs the dynamics of the asymmetry parameter.
When σλ approaches zero, λt becomes static and assumes the value of its initial point, λ0 . Furthermore, if the initial point is also zero, the model reduces to the vanilla SV model. Hence, by inducing sparsity for both σλ and λ0 , we can encompass all three cases of interest.”
2) Improved Section 4
Section 4 was heavily revised in order to improve the paper. We opted to focus on the main differences between HMC and RWMH and the reasons that lead us to choose HMC over RWMH. Additionally, we also added Turkman et al. [2019] as an additional textbook reference for both RWMH and HMC.

Minor comments:
1) Figure 1:
We changed Figure 1 to have different line types (solid, dashed and dotted). We added an explanation about how each line type corresponds to a different λ parameter on the graph’s caption.
2) Difference between equation 1 and 11.
Since εt ∼ N (0, 1), Equation (1) implies that yt follows a normal with a location of 0 and a scale of exp(ht /2). By replacing εt with zt ∼ SN (λt ), Equation (11) implies that yt follows a skew-normal with a location of 0, a scale of exp(ht /2) and a skewness parameter λ t . Therefore, Equation 11 comes from a (potentially) skewed distribution while Equation 11 implies a symmetric distribution.
We highlight this difference by adding the following phrase when explaining Equation 11:
” Since εt ∼ N (0, 1), Equation (1) implies that yt follows a normal with a location of 0 and a scale of exp(ht /2). By replacing εt with zt ∼ SN (λt ), Equation (11) implies that yt follows a skew-normal with a location of 0, a scale of exp(ht /2) and a skewness parameter lambdat . Therefore, Equation 11 comes from a (potentially) skewed distribution while Equation 11 implies a symmetric distribution. ”
Additionally, note that we didn’t alter the specification for ht . The only modification is that we replaced εt by zt .
3) Bayes factors vs BIC:
No. We meant Bayes factor. Bayes factors can be used to compare models as described in Turkman, Paulino and Müller’s textbook subsection 5.2.3 entitled ”Model Selection Using Bayes Factors”. We address this comment by adding Turkman, Paulino and Müller’s textbook as a reference when commenting on model comparison via Bayes factor in Section 3.

Reviewer 4 Report

Comments and Suggestions for Authors

The paper expands traditional stochastic volatility models by allowing for time-varying skewness. The results are relevant to the literature on stochastic volatility. Although the paper is generally well written and clear, I have the following comments:

- Section 5: The authors focus their analysis on the third moment and mention that this is much less explored than the first two moments. However, no references are provided for some studies focusing on the third moment (except for one in the introduction). Further references would be appreciated, together with a short paragraph with a summary of what was done and the results obtained.

- Please provide further insights about the choice of the priors presented in Appendices B and C. Why these and not others? How large would be the impact of choosing other priors on the results?

- How does your model behave when compared with other models available in the literature (frequentist and Bayesian; that consider time-varying skewness or not)? For instance, analyzing Table 3 without comparing it with other methods, it is not clear whether the results are good or not)

- Figure A1 should be mentioned in the text

- Aiming at reproducible science, I strongly suggest the authors to provide a link to the repository with the data and the code to obtain the results presented in the paper.

Comments on the Quality of English Language

- The English should be carefully revised

Author Response

Dear, reviewer.
Thank you for your comments. We addressed all your points and made several changes to the original text in order to improve the paper.
1) Section 5:
We improve Section 5 by providing additional references focusing on the third moment.
” The bond market is one of the largest in the world being key for investors and policy makers. Most papers focus on the first two moments e.g. Litterman and Scheinkman [1991], Collin-Dufresne and Goldstein [2002], Cochrane and Piazzesi [2005] and Joslin [2018]. Our paper focus on the much less explored third  moment. Skewness captures the likely direction of returns allowing an interest rate investor to improve their forecast about the sign of future yields. Such feature was explore in the literature in at least three ways. First, Bianchi et al. [2022] highlights the role of skewness in forecasting crashes in momentum portfolios. Second, both Bauer and Swanson [2023] and Cieslak and McMahon [2023] highlight that even monetary ’surprises,’ such as the differences between actual policy rates announced by the Federal Open Market Committee (FOMC) and expectations from professional forecasters, can be partially predicted by the option-implied skewness of the 10-year US bond. Third, Li and Scharth [2020]
demonstrate that allowing for skewness improves value-at-risk evaluation for multiple assets. ”
We included a short paragraph describing the main results of the section and how they were obtained. ”In our first application, we model the monthly yield changes in fixed 1-year maturity for both American and Brazilian bonds. We sample from the joint posterior via the HMC scheme presented in Section 3 by combining the likelihood implied by our proposed model in Section 2 and the priors described in Appendix B. In both cases, skewness is likely to be time-varying. It is associated with cycles of monetary easing and tightening. It is partially explained by the central bank’s mandate and provides valuable information regarding the future direction of yield changes.”
2) Priors
Our prior represent uninformative beliefs while also respecting non-negative restrictions for scale parameters. We added insights about the choice of priors on Appendixes B and C.
3) Table 3 comparison with other models
For the bond yield application, we added a comparison with the generalized autoregressive score (GAS) model. Similar to our proposal, the GAS model also allows for time-varying skewness and time-varying volatility. In all
cases considered in Section 4, our model predicts the likely direction of yield changes at least as well as GAS models with clear signs of improvement when focusing of the Brazilian case.
4) Figure A1
We included the following text before mentioning Figure A1:
”Also, Figure A1 plots the skewness itself using green and red shades to indicate easing and tightening periods in addition to the posterior for {λt}. While the conclusions are the same, the version with skewness may help the reader to make a better assessment of the magnitude of skewness in each period."

5) Data and code
All codes and data to reproduce the results of the paper will be made available by the first author on his Github, github.com/igorfbmartins , shortly after publication. We added a footnote to the text to inform the reader

Round 2

Reviewer 3 Report

Comments and Suggestions for Authors

I appreciate the effort in improving the manuscript and in answering my questions. The paper is now much easier to read, and the contributions are nice and clear.

Comments on the Quality of English Language

Although the manuscript is fine as it is, I would suggest one more read-through, as I noticed some small errors and typos. 

Reviewer 4 Report

Comments and Suggestions for Authors

The authors responded to all my concerns and made the necessary changes in the manuscript.